# IL-21 (rs2055979 and rs2221903)/*IL-21R* (rs3093301) Polymorphism and High Levels of IL-21 Are Associated with Rheumatoid Arthritis in Mexican Patients

**DOI:** 10.3390/genes14040878

**Published:** 2023-04-07

**Authors:** Noemi Magdalena Carreño-Saavedra, Itzel Viridiana Reyes-Pérez, Andrea Carolina Machado-Sulbaran, Gloria Esther Martínez-Bonilla, María Guadalupe Ramírez-Dueñas, José Francisco Muñoz-Valle, Valeria Olaya-Valdiviezo, Trinidad García-Iglesias, Erika Aurora Martínez-García, Pedro Ernesto Sánchez-Hernández

**Affiliations:** 1Laboratorio de Inmunología, Departamento de Fisiología, Centro Universitario de Ciencias de la Salud (CUCS), Universidad de Guadalajara (UDG), Guadalajara 44340, Jalisco, Mexicoitzelreyesp15@hotmail.com (I.V.R.-P.); andrecaroms@gmail.com (A.C.M.-S.); valolayaval@gmail.com (V.O.-V.); trini.iglesias@gmail.com (T.G.-I.); 2Doctorado en Ciencias Biomédicas, Departamento de Fisiología, Centro Universitario de Ciencias de la Salud (CUCS), Universidad de Guadalajara (UDG), Guadalajara 44340, Jalisco, Mexico; 3Instituto de Investigación en Ciencias Biomédicas, Departamento de Clínicas Médicas, Centro Universitario de Ciencias de la Salud (CUCS), Universidad de Guadalajara (UDG), Guadalajara 44340, Jalisco, Mexico; biologiamolecular@hotmail.com; 4Cuerpo Académico UDG-CA-1135 “Biomarcadores Inmunogenéticos y Factores Farmacológicos en Enfermedades Crónico-Degenerativas”, Centro Universitario de Ciencias de la Salud (CUCS), Universidad de Guadalajara (UDG), Guadalajara 44340, Jalisco, Mexico; 5Instituto de Investigación en Cáncer en la Infancia y Adolescencia, Centro Universitario de Ciencias de la Salud (CUCS), Universidad de Guadalajara (UDG), Guadalajara 44340, Jalisco, Mexico; 6Servicio de Reumatología, Hospital Civil Fray Antonio Alcalde, Guadalajara 44280, Jalisco, Mexico; glomarbon@hotmail.com; 7Departamento de Biología Molecular y Genómica, Instituto de Investigación en Reumatología y del Sistema Músculo Esquelético (IIRSME), Centro Universitario de Ciencias de la Salud (CUCS), Universidad de Guadalajara (UDG), Guadalajara 44340, Jalisco, Mexico

**Keywords:** rheumatoid arthritis, SNP, polymorphism, IL-21, IL-21R, biomarkers

## Abstract

Rheumatoid Arthritis (RA) is characterized by joint destruction, chronic inflammation, and autoantibody production. IL-21/IL-21R plays an essential role in the immunopathology of RA. Elevated IL-21 serum levels have been associated with RA and disease activity. Here, we evaluated the association of *IL-21*/*IL-21R* polymorphisms and IL-21 serum levels with RA. The study included 275 RA patients and 280 Control subjects (CSs). Single nucleotide polymorphisms *IL-21* (rs2055979 and rs2221903) and *IL-21R* (rs3093301) were genotyped using PCR-RFLP. Clinical activity was evaluated by DAS28-ESR; IL-21 and anti-CCP serum levels were quantified by ELISA. The *IL-21* rs2055979 AA genotype was higher in RA patients than in the CS group (*p* = 0.0216, OR = 1.761, 95% CI = 1.085–2.859); furthermore, RA patients showed anti-CCP elevated levels compared to the CA genotype (*p* = 0.0296). The *IL21R* rs3093301 AA genotype was also higher in RA patients than in the CS group (*p* = 0.0122, OR = 1.965, 95% CI = 1.153–3.348). The AT haplotypes of *IL-21* rs2055979 and rs2221903 were more frequent (49%) in the RA group (*p* = 0.006). IL-21 serum levels were significantly elevated in the RA group, but without an association with *IL-21* polymorphisms. In conclusion, *IL-21* rs2255979 and *IL-21R* rs3093301 are associated with a higher risk of RA, and could be a genetic marker. Moreover, the elevated IL-21 levels in RA suggest that IL-21/IL-21R could be a therapeutic target in RA.

## 1. Introduction

Rheumatoid Arthritis (RA) is a multifactorial disease characterized by joint destruction and chronic inflammation, autoantibody production (rheumatoid factor [RF] and anti-citrullinated protein antibody [ACPA]), and systemic features. The disease is associated with irreversible damage, and progressive disability leads to a decreased quality of life [1,2].

In RA pathogenesis, proinflammatory cytokines play a significant role in inducing an inflammatory response and the release of other inflammation mediators. In this context, the participation of interleukin-21(IL-21)/IL-21R (IL-21 receptor) plays an essential role in the disease activity, which is associated with elevated serum levels of IL-21 in RA patients [3,4]. 

IL-21 is a class I cytokine produced by activated CD4+ T cells, natural killer T (NKT) cells, and T helper (Th) cells [5]. IL-21 has a variety of effects on the immune system and plays a critical role in B-cell responsiveness, proliferation, plasma differentiation, and immunoglobulin production; thus, the IL-21 effects on B cells may contribute to the development of autoimmune diseases [6]. The receptor of IL-21 (IL-21R) is a heterodimer with an α chain and a common γ chain subunit. It is expressed mainly in immune cells, such as T and B lymphocytes, NK cells, macrophages, and dendritic cells, as well as in some non-immune cells, such as intestinal epithelial cells, fibroblasts, keratinocytes, and endothelial cells [5,7]. 

Molecular signaling occurs through the JAK-STAT, PI3K, and MAPK pathways, inducing STAT3 activation, which is crucial for B- and T-cell differentiation [8]. STAT3 plays a relevant role in Th17 cell differentiation and proliferation. Recent studies on RA showed that Th17 survival is through STAT3, dependent on the IL21/IL-21R interaction [7]. Through signaling pathways activated by IL-21R on fibroblast-like synoviocytes (FLS), IL-21 favors its proliferation and survival, increases migration and invasion, and promotes the expression of MMP-3 and MMP-9, relevant for the degradation of extracellular matrix proteins [7,9].

The *IL-21* gene is located on chromosome 4q27 and has five exons, while the *IL-21R* gene is located on chromosome 16p12.1 and has 11 exons. Previous investigations reported the association of diverse *IL-21* and *IL-21R* polymorphisms with autoimmune disorders, such as systemic lupus erythematosus (SLE), multiple sclerosis, autoimmune thyroid diseases, and RA, in different populations [3,10,11,12]. *IL-21* rs2055979 and rs2221903 are located in the second intron, and *IL-21R* rs3093301 is located in the second intron. Due to the involvement of IL-21 in autoimmune diseases, it is possible that these polymorphisms are associated with RA development and IL-21 levels. This study aimed to evaluate the association of *IL-21*/*IL21R* polymorphisms and IL-21 serum levels with RA, as well as acute phase reactants, autoantibodies, and a possible clinical implication for the Mexican population.

## 2. Materials and Methods

### 2.1. Subjects

A total of 555 subjects were recruited, including 275 RA patients (RA) from the Rheumatology Department of the Hospital Civil de Guadalajara “Fray Antonio Alcalde”, who were classified according to the American College of Rheumatology/European League Against Rheumatism (ACR-EULAR) 2010 criteria. The exclusion criteria were: coexistence of other autoimmune diseases, uncontrolled systemic diseases or malignancies, treatment with corticosteroids at doses greater than 15 mg per week, pregnancy, and presence of inflammatory or infectious diseases at the time of blood sample collection. The control subject group (CS) consisted of 280 subjects who were randomly selected, clinically healthy, and without a family history of autoimmune diseases. All RA patients and CSs were from western Mexico, at least until the third generation; over 18 years of age; and signed informed consent forms. The biosafety, ethics, and research committees of the Hospital Civil de Guadalajara “Fray Antonio Alcalde” approved this study (Reg. No.106/18).

### 2.2. Clinical Assessment

Patients were evaluated by a rheumatologist. Clinical activity was evaluated by the disease activity score 28 (DAS28)-erythrocyte sedimentation rate (ESR) (DAS28-ESR), and was classified as high, moderate, low, or remission for each patient. The RA patients were stratified into three groups according to the disease evolution: very early RA (<3 months), early RA (<1 year), and established RA (>1 year).

### 2.3. Inflammatory and Serological Parameters

Peripheral blood samples were obtained from all individuals included in the study. Serum was separated by centrifugation and stored at −70 °C until use. Erythrocyte sedimentation rate (ESR) was performed using the Wintrobe method (reported in mm/h), and C-reactive protein (CRP) was assessed by turbidimetric assay (BioSystems, Barcelona, Spain) (reported in mg/L, detection limit 6 mg/L). RF (isotype IgM) detection was performed by turbidimetric assay (BioSystems, Barcelona, Spain) on automatic equipment; the cut-off value for RF positivity was 30 IU/mL (detection limit 2 IU/mL). The IgG Anti-cyclic citrullinated peptides (anti-CCP antibodies) were quantified by an ELISA kit (Axis-Shield Diagnostics Limited, Dundee, Scotland); the cut-off value for anti-CCP positivity was 5 U/mL (detection limit 1.04 U/mL).

### 2.4. Genomic DNA Extraction and Genotyping

Total genomic DNA (gDNA) was extracted from EDTA-anticoagulated peripheral blood samples of both RA patients and CSs using the modified Miller’s method [13]. 

*IL-21* and *IL-21R* polymorphism genotyping were carried out using the polymerase chain reaction-restriction fragment length polymorphism (PCR-RFLP) method. 

The *IL-21* rs2055979 (C/A) polymorphism was genotyped using the primers 5′-CAGCCAGGAAACTCTGGAAAGAA-3′ (upstream) and 5′-GCTCTGAACCCAAACACTCTCATTT-3 (downstream) in order to amplify a 212-bp PCR fragment. 

The *IL-21* rs2221903 (T/C) polymorphism was genotyped using the primers 5′-TGGACACTGACGCCCATATTGA-3′ (upstream) and 5′- AAGGCAGTTTAGTGGCGACAGCT-3 (downstream) in order to amplify a 230-bp PCR fragment. 

The *IL-21R* rs3093301 (T/A) polymorphism was genotyped using the primers 5′-CCCTCCCTCTTTCTTTGTTAG-3′ (upstream) and 5′-TCCTCCTACCTCGGCCTCTCAAAGTG-3′ (downstream) in order to amplify a 182-bp PCR fragment.

The PCR reaction for each *IL-21* and *IL-21R* polymorphism was performed in a total volume of 25 µL containing 200 ng of gDNA, primers at 0.75 µM for rs2055979, 0.375 µM for rs2221903, and 0.075 µM for rs3093301, and 10 µL of BioMixTM Red (Bioline). PCR reactions for *IL-21* rs2055979 and *IL-21* rs2221903 were carried out with the following conditions: the initial denaturation at 94 °C for 5 min, followed by 35 cycles at 94 °C for 50 s, 62 °C for 50 s, and a final extension at 72 °C for 5 min. For *IL-21R* rs3093301, these were 58 °C for 30 s and 72 °C for 30 s, with a final extension at 72 °C for 5 min. The amplicons were digested with restriction enzymes incubating for 16 h at 37 °C in a water bath, and the RFLP products were analyzed with 8% bisacrylamide gel. The genotypes of each polymorphism were determined according to the digestion patterns. The *NlaIII* was used for *IL-21* rs2055979 (CC, 158 bp, and 54 bp; CA, 212 bp, 158 bp, and 54 bp; and AA, 212 bp) and *IL-21R* rs3093301 (TT, 143 bp, and 39 bp; TA, 182 bp, 143 bp, and 39 bp; and AA, 182 bp). The *MboII* was used for *IL-21* rs2221903 (CC, 149 bp, and 81 bp; TC, 230 bp, 149 bp, and 81 bp; and TT, 230 bp).

### 2.5. Cytokine Quantification

IL-21 serum levels of the 275 RA patients and 242 CSs were determined using an enzyme-linked immunosorbent assay (ELISA) kit (detection limit 16 pg/mL) according to the manufacturer’s instructions (BioLegend, San Diego, CA, USA). The optical density of samples was read using a microplate reader (Synergy HT, Biotek, Winooski, VT, USA) at 450 nm. The IL-21 concentration was calculated using a standard curve fitted with sigmoidal four-parameter logistic (4PL) regression, and the values outside of the ELISA sensitivity range were omitted for statistical analyses.

### 2.6. Statistical Analysis

Descriptive statistics were used for clinical data. The Kruskal–Wallis and Mann–Whitney U tests were used for the comparison of non-parametric data. Chi-square (χ2) and Fisher’s exact tests were used to compare genotype and allele frequencies between the study groups. A *p* < 0.05 was considered as statically significant. All of the statistical analyses were performed using the GraphPad Prism 8 software. The Hardy–Weinberg equilibrium test, as well as the analysis of haplotype and the linkage disequilibrium between the polymorphisms were completed by SHesis software (http://analysis.bio-x.cn/myAnalysis.php, accessed on 18 January 2023). 

## 3. Results

### 3.1. Demographics and Clinical Characteristics

The RA group consisted of 225 (81.8%) females and 50 (18.2%) males, with a mean age of 48.36 ± 12.65 years. The CS group consisted of 222 (79.3%) females and 58 (20.7%) males, with a mean age of 36.06 ± 12.53 years. ESR and CRP levels were significantly elevated in RA patients compared to those in the CS group (*p* < 0.0001). 

Regarding the disease stage, 78.6% patients were in the established phase, 17.8% early, and 3.6% very early. In relation to disease activity, 44% had moderate activity. About 72.7% of patients were high positive for anti-CCP (Figure 1) and 26.9% were high positive for RF. Regarding disease-modifying antirheumatic drugs (DMARDs), 86.55% were under treatment, of which 28.99% had monotherapy, 39.92% double therapy, and 31.09% triple therapy. At the time of inclusion, 13.45% of the patients had no treatment (Table 1).

### 3.2. Association of the IL-21/IL-21R Polymorphisms with RA

We genotyped *IL-21* (rs2055979 and rs2221903), and *IL-21R* (rs3093301) polymorphisms in 275 RA patients and 280 CSs (Table 2). The alleles and genotype distributions of the *IL-21* and *IL-21R* polymorphisms were in Hardy–Weinberg equilibrium (*p* > 0.05). 

The frequency of homozygous (AA) *IL-21* rs2055979 polymorphism was higher in RA patients than in CSs (*p* = 0.0216, OR = 1.761, 95% CI = 1.085–2.859). Furthermore, the allele A was more frequent in RA patients compared to CSs (*p* = 0.0145, OR = 1.343, 95% CI = 1.060–1.702), similar to the dominant genetic model (*p* = 0.0057, OR = 1.686, 95% CI = 1.162–2.445).

We did not find any statistical difference between RA patients and CSs in *IL-21* rs2221903. 

Concerning the *IL-21R* rs3093301, the genotype AA (*p* = 0.0122, OR = 1.965, 95% CI = 1.153–3.348) and allele A (*p* = 0.0418, OR = 1.289, 95% CI = 1.009–1.646) were higher in RA patients than in CSs, similar to the recessive genetic model of the A allele (*p* = 0.0077, OR = 0.512, 95% CI = 0.312–0.843).

### 3.3. Haplotype Frequencies of IL-21 SNPs and the Association with RA

We analyzed the haplotypes of the *IL-21* SNPs (rs2055979 and rs2221903). The analysis showed that these SNPs are in linkage disequilibrium (D′ = 0.86, r^2^ = 0.098) Figure 2. We found that the AT haplotype was more frequent (49%) in the RA group (AT, *p* = 0.0060, OR = 1.39; 95% CI = 1.099–1.769), and the CT haplotype was more frequent in the CS group (46%) (CT, *p* = 0.0009, OR 0.66, 95% CI = 0.524–0.848) (Table 3). 

### 3.4. Association of IL-21 and IL-21R Polymorphisms with DAS28-ESR, Acute Phase Reactants, and Autoantibody Positivity

Considering the genotypes, both disease activity and serological parameters were analyzed in order to assess the involvement of polymorphisms in RA (Table 4). The analysis of DAS28-ESR and acute phase reactants (ESR and CRP) with the *IL-21* and *IL-21R* polymorphisms did not show differences between the different genotypes.

### 3.5. Autoantibody Levels

The *IL-21* rs2055979 polymorphisms showed that carriers of the AA genotype have higher levels of anti-CCP than the CA genotype (*p* = 0.0296). The *IL-21* rs2221903 and *IL-21R* rs3093301 genotypes did not show an association with anti-CCP levels. The *IL-21* rs2221903 TT genotype showed higher levels of RF (*p* = 0.0292). The *IL-21* rs2055979 and *IL-21R* rs3093301 genotypes did not show differences in RF levels (Table 4). 

### 3.6. Levels of IL-21

The IL-21 serum levels were higher in RA patients than in the CS group (*p* < 0.0001) (Figure 3). However, comparing IL-21 levels between genotypes of the *IL-21*/*IL21R* polymorphisms did not provide evidence of differences (Table 4). No significant differences in IL-21 levels were observed between patients with or without treatment (*p* = 0.067). In contrast, elevated IL-21 levels were observed in anti-CCP-positive (*p* = 0.0007) and RF-positive (*p* < 0.0001) patients, and in those positive for both autoantibodies (*p* < 0.002). Additionally, higher IL-21 levels were observed in patients with active disease compared to those in remission (*p* = 0.013), and in patients with established disease, although this did not show statistical significance (*p* = 0.4288).

### 3.7. Combined Effect of IL-21 Polymorphisms on IL-21 Serum Levels

We analyzed the combined effect of haplotypes of *IL-21* polymorphism with the serological parameters, finding that CATT carriers have higher RF levels than CATC carriers (*p* = 0.0128). The rest of the analyses did not show any association (Table 5). 

## 4. Discussion

In RA development, different cytokines have an essential role in the inflammatory environment, leading to a lack of control in the cellular immune response, and abnormal autoantibody production [14,15]. IL-21 is one of the main cytokines involved, and it plays a central role in different cell types, including both immune cells and non-immune cells; among its functions are to induce proliferation and survival of T CD4+ cells, and to secrete proinflammatory cytokines for Th17 differentiation [7,16]. 

RA has a strong genetic component, within *HLA-DRB1* alleles, and genes such as *PADI4*, *PTPN22,* and *CTLA4,* which have been associated with the pathology. Polymorphisms in cytokine and/or cytokine receptor genes that increase the risk for RA development have also been reported [17,18]. *IL-21*/*IL21R* polymorphisms have also been reported to be associated with risk for autoimmune diseases such as multiple sclerosis, autoimmune thyroid diseases, SLE, and RA [11,12,19,20].

In this study, we found an association of the *IL-21* rs2055979 (C/A) polymorphism with RA, in that the subjects carrying the allele A had a 1.3-fold higher susceptibility to RA, compared to carriers of allele C; likewise, the AA genotype carriers increased to 1.7-fold susceptibility. Similar to our results, a study in Chinese RA patients reported that allele A carriers are 2.1-fold more likely to develop RA, and AA genotype carriers are 4.3-fold more likely. Another study in the Chinese population also reported that allele A carriers have a 1.4-fold higher risk of developing SLE [11,20]. 

On the other hand, the *IL-21* rs2221903 (C/T) polymorphism has been associated with different autoimmune diseases; Zhang et al., in 2013, found an association of the T allele with the development of autoimmune thyroid disease in the Chinese population [12]. Malinowski et al., in 2017, reported that the CT and CC genotypes are associated with active RA in the Caucasian population [3]. In our population, we have not observed this association in rs2221903, and this could be explained by the small sample size of subjects with the CC genotype in our population. 

The *IL-21R* rs3093301 (T/A) polymorphism has been described in different infectious diseases; however, few studies have associated it with autoimmune diseases. A previous study in the Chinese population reports an association of the genotypes TA and TT *IL-21R* rs3093301 with the development of autoimmune thyroid disease [12]. Additionally, the presence of the A allele was associated with a higher susceptibility to developing SLE [21]. We found that A allele carriers had a 1.2-fold higher susceptibility to RA, and carriers of the AA genotype increased susceptibility to 1.9-fold higher. 

In order to assess whether the SNPs are directly related to the disease, we analyzed whether they are related to disease activity; however, no differences were found between SNPs and the DAS28-ESR score. In contrast, Hao et al., in 2021, observed a significant association between *IL-21* rs2055979 polymorphism and the DAS28 score: the subjects with the AA genotype had higher DAS28 scores than those with the CA or CC genotypes [20]. Malinowsky et al., in 2017, classified patients into remission with DAS28 < 2.5, and active disease with DAS28 > 2.5, and found that among patients with DAS28 > 2.5, there was an increased prevalence of rs2221903 CT and CC genotypes [3]. However, the discrepancy between our results and these other studies could be due to differences in the type of population (Chinese and Polish, respectively, in the aforementioned studies), and because our patients comprise a heterogeneous group in evolution time, disease activity, and treatments used.

The SNPs studied are intronic variants *IL-21* rs2055979 (intron 2), *IL-21* rs2221903 (intron 2), and *IL-21R* rs3093301 (intron 2) (https://www.ncbi.nlm.nih.gov/snp/). The function of intronic SNPs is not yet fully understood; however, it has been proposed that it could be related to controlling gene expression, by regulating mRNA splicing and translation [22]. 

We examined IL-21 levels in RA patients and the CS group, and assessed whether the levels are related to SNPs. First, we found elevated levels in RA patients compared to those in the CS group, similar to a previous study, also in the Mexican population [4]. Furthermore, Chinese RA patients showed higher IL-21 levels than healthy controls [20]. In addition, a previous report on SLE patients showed increased IL-21 levels compared to control subjects [11]. This information suggests a possible role of IL-21, as a strong promoter of inflammation, in developing the pathogenesis of autoimmune diseases such as SLE and RA [23]. 

Moreover, IL-21 promotes the proliferation of B cells and memory B cells, as well as plasma cell differentiation in collaboration with Tfh (T follicular helper) cells [24]. Secondly, we found elevated IL-21 serum levels in RA patients compared to those in the CS group; in addition, the highest IL-21 levels were observed in anti-CCP-positive and RF-positive patients, or positives for both autoantibodies. The presence of autoantibodies is a significant characteristic in RA patients; the majority of our patients were positive to anti-CCP antibodies. In autoimmune disease, the autoantigen accumulation and potent triggers can attract and activate dendritic cells and B cells for autoantibody production. In seropositive RA patients, the anti-CCP—which is a type of ACPA—and RF are the most frequent, commonly co-occur, and could play a role in RA immunopathogenesis [25]. In addition to finding high levels of IL-21 in RA patients, these levels have been associated with high RF and anti-CCP titers, as well as with seropositive patients to both, according to the cut-off points [4]. The increase in IL-21 levels in autoimmune diseases, and its association with RA clinical manifestations, such as the relationship with autoantibodies, suggests a regulation mechanism via IL-21/IL-21R [7].

Regarding SNPs, in our current research findings, RA subjects carrying the genotype AA of *IL-21* rs2055979 had an association with higher anti-CCP levels. It has been described that IL-21 promotes B-cell proliferation and plasma cell differentiation, which is crucial for antibody production [26]. In this case, the SNP could be favoring the production of antibodies; however, an explicit association has not been described. 

IL-21 serum levels were evaluated according to *IL-21* and *IL-21R* genotypes, and no difference was found between them, which is not in agreement with Hao et al., who reported in 2021 that the *IL-21* rs2055979 AA genotype in RA patients showed elevated IL-21 levels compared to other genotypes [20]. Additionally, Lan et al., in 2014, reported elevated IL-21 levels in SLE patients with AA and CA genotypes. Both studies were conducted in the Chinese population [11]. A possible explanation is that the population is genetically different, with exposure to several environmental factors; in addition, we have a very heterogeneous RA, in terms of evolution time and the variety of treatments.

In this study, in the haplotype analysis of *IL-21* SNPs (rs2055979/rs2221903), we found that subjects carrying the AT haplotype had a 1.39-fold higher susceptibility to RA; in contrast, carriers of the CT haplotype have a minor risk (OR 0.66) of developing RA. Moreover, analysis of the combined effect of the genotypes showed an association between CATC, with higher levels of RF compared to CATT. 

Our results show that the *IL-21* rs2055979 A allele is a risk for developing RA; moreover, the AA genotype poses a risk of both developing RA and showing high levels of anti-CCP. *IL-21* rs2221903 did not show differences between RA patients and the CS group, but TT showed elevated levels of RF. The *IL-21R* rs3093301 A allele and AA genotype pose a risk of RA, unrelated to autoantibody levels. The rs2055979/rs2221903 AT haplotype also poses a higher risk of RA, suggesting that these SNPs, mainly *IL-21* rs2055979 polymorphism, are genetic markers of RA susceptibility, and of poor prognostic factors such as anti-CCP, but they are not related to the increased IL-21 levels observed in RA patients. Genetic variables such as polymorphisms, and biological markers such as cytokines, have been proposed as predictors of disease and response to treatment in RA; however, the results have not been consistent [27]. 

Several SNPs, in genes such as *DHFR*, *MTHFR,* and *RFC1,* related to the MTX pathway have been evaluated as predictors of treatment response, so in the future, other SNPs related to IL-21 and their relationship with treatment response could also be researched [28,29]. In addition, IL-21R expression studies could provide information on the mechanisms in which IL-21 is associated with the activation of the immune system in patients with these polymorphisms. 

## 5. Conclusions

In conclusion, our study confirms that *IL-21* rs2255979 and rs3093301 are associated with a higher risk of RA, without association with IL-21 elevated levels; moreover, *IL-21* rs2055979 AA genotype carriers are more susceptible to having higher levels of anti-CCP. This indicates that *IL-21* and *IL-21R* SNP teamwork is related to the development of RA, and could be a genetic marker. Moreover, the elevated IL-21 levels in RA suggest that IL-21, or IL-21R, could be a therapeutic target.

## Figures and Tables

**Figure 1 genes-14-00878-f001:**
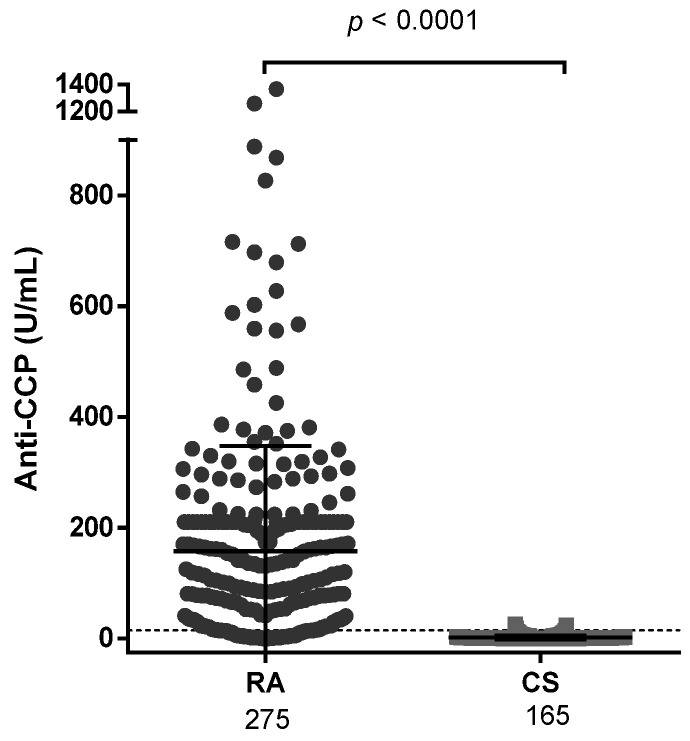
Anti-CCP levels in RA patients (*n* = 275) and CSs (*n* = 165); the dotted line represents the value to define high positives. Data are presented as anti-CCP U/mL mean ± SD and were compared by Mann–Whitney U test.

**Figure 2 genes-14-00878-f002:**
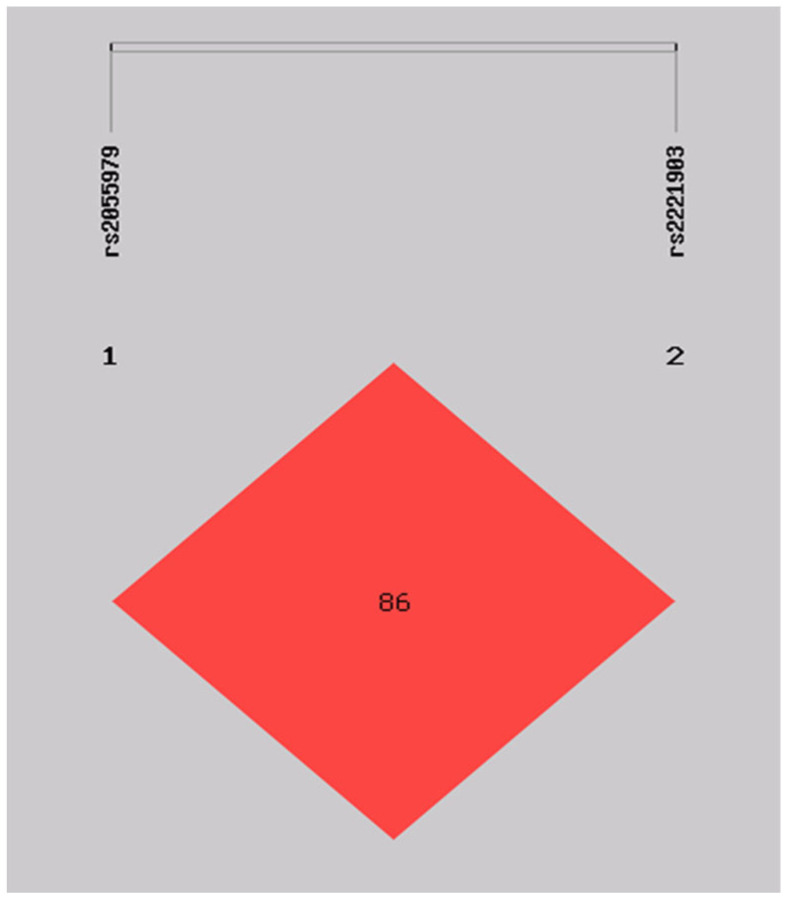
Analysis of *IL-21* SNPs (rs2055979 and rs2221903). Linkage disequilibrium (D′ = 0.86, r^2^ = 0.098).

**Figure 3 genes-14-00878-f003:**
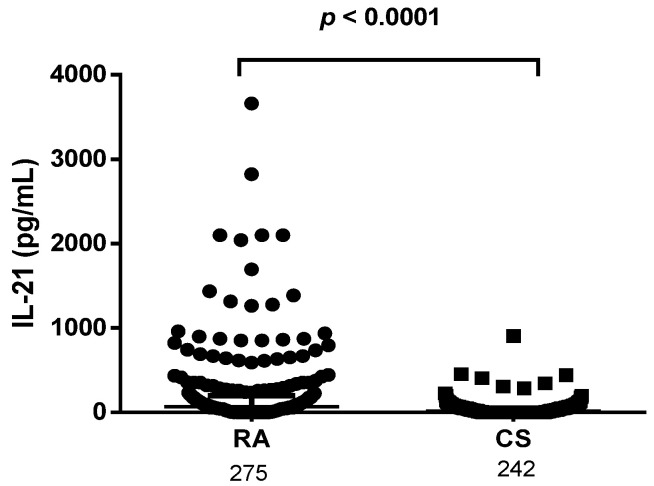
IL-21 serum levels in RA patients (*n* = 275) and CSs (*n* = 242). Data are presented as IL-21 pg/mL mean ± SD and were compared by Mann–Whitney U test.

**Table 1 genes-14-00878-t001:** Demographic and clinical features in RA patients and CSs.

Parameters	RA *n* = 275	CS *n* = 280	*p*
Gender (F/M)	225/50	222/58	
Age (years) *	48.36 ± 12.65	36.06 ± 12.53	
Disease evolution (years) *	7.1 ± 7.9		
Serological parameters			
ESR (mm/h) *	35.15 ± 15.72	18.51 ± 10.87	<0.0001
CRP (U/mL) *	19.28 ± 25.36	4.12 ± 4.28	<0.0001
RF (UI/mL) *	61.42 ± 27.04	6.87 ± 8.28	<0.0001
Anti-CCP (U/mL) *	157.50 ± 190.30	1.94 ± 3.65	<0.0001
Stages of disease	% (*n*)		
Established > 1 year	78.6 (216)	_	
Early < 1 year	17.8 (49)	_	
Very early < 3 months	3.6 (10)	_	
Disease activity (DAS28-ESR)	% (*n*)		
High	20.6 (51)	_	
Moderate	44.0 (109)	_	
Low	16.9 (42)	_	
Remission	18.5 (46)	_	
Risk Factors	% (*n*)		
Smoking	32.0 (88)	_	
Extra-articular manifestations	24.0 (66)	_	
High RF positives ^a^	26.9 (74)	_	
High anti-CCP positives ^a^	72.7 (200)	_	
Treatment	% (*n*)	Treatment combination	
Without treatment	13.45 (37)		
With treatment	86.55 (238)		
Monotherapy	28.99 (69)	MTX/LEF/SSZ/CQ	
Double therapy	39.92 (95)	MTX-SSZ/MTX-CQ/CQ-SSZ	
Triple therapy	31.09 (74)	MTX-CQ-SSZ	

Clinical-demographic data are shown. RA: Rheumatoid arthritis; CS: control subjects. ESR: erythrocyte sedimentation rate; CRP: C-reactive protein; RF: rheumatoid factor; Anti-CCP: anti-cyclic citrullinated peptide antibodies; DAS28-ESR: Disease Activity Score 28- erythrocyte sedimentation rate. Treatment frequencies and doses are shown. MTX: methotrexate; SSZ: sulfasalazine; CQ: chloroquine; LEF: leflunomide. ^a^ Autoantibodies were classified as low positive (LP) when <3 times the cut-off point and high positive (HP) when >3 times the cut-off point, according to the criteria established by ACR/EULAR. * Data are presented in mean with standard deviation. The analysis was carried out by Mann–Whitney U test.

**Table 2 genes-14-00878-t002:** Genotype and allele frequencies of *IL-21*/*IL-21R* polymorphism in RA patients and CSs.

Polymorphism	RA	CS	*p*	OR (95% CI)
	N = 275 (%)	N = 280 (%)		
*IL-21* rs2055979 C/A				
Genotypes				
CC	65 (23.64)	96 (34.29)		
CA	148 (53.82)	132 (47.14)	**0.0115**	1.656 (1.118–2.452)
AA	62 (22.55)	52 (18.57)	**0.0216**	1.761 (1.085–2.859)
Alleles				
C	278 (51)	324 (58)		
A	272 (49)	236 (42)	**0.0145**	1.343 (1.060–1.702)
Model				
Recessive CC + CA AA	213 (77.45) 62 (22.54)	228 (81.42) 52 (18.57)	0.2466	0.784 (0.518–1.184)
Dominant AA + CA CC	210 (76.36) 65 (23.64)	184 (65.71) 96 (34.29)	**0.0057**	1.686 (1.162–2.445)
*IL-21* rs2221903 T/C				
Genotypes				
TT	203 (73.82)	214 (76.43)		
TC	70 (25.45)	58 (20.71)	0.2345	1.272 (0.855–1.893)
CC	2 (0.73)	8 (2.86)	0.0728	0.264 (0.55–1.256)
Alleles				
T	476 (86.5)	486 (86.79)		
C	74 (13.4)	74 (13.21)	0.9062	1.021 (0.722–1.443)
Model				
Recessive TT + CT CC	273 (99.27) 2 (0.71)	272 (97.14) 8 (2.85)	0.0592	4.015 (0.845–19.078)
Dominant CC + CT TT	72 (26.18) 203 (73.81)	66 (23.57) 214 (76.42)	0.4768	1.150 (0.782–1.691)
*IL-21R* rs3093301 T/A				
Genotypes				
TT	106 (38.55)	119 (42.5)		
TA	120 (43.64)	133 (47.5)	0.9442	1.013 (0.707–1.451)
AA	49 (17.82)	28 (10)	**0.0122**	1.965 (1.153–3.348)
Alleles				
T	332 (60.36)	371 (66.25)		
A	218 (39.64)	189 (33.75)	**0.0418**	1.289 (1.009–1.646)
Model				
Recessive TT + TA AA	226 (41.09) 49 (8.90)	252 (90) 28 (10)	**0.0077**	0.512 (0.312–0.843)
Dominant TA + AA TT	169 (61.45) 106 (38.54)	161 (57.5) 119 (42.5)	0.3427	1.178 (0.839–1.654)

Frequency distributions of *IL-21/IL21* polymorphisms. RA: rheumatoid arthritis; CS: control subjects; Chi-square test. OR: Odds Ratio; 95% Confidence Intervals. The bold letters indicate significant differences.

**Table 3 genes-14-00878-t003:** Haplotype distribution in RA patients and Control Subjects.

IL-21 (rs2055979/rs2221903) Haplotype	RA 2n (%)	CS 2n (%)	OR (CI 95%)	*p*
AT	271.98 (0.495)	227.31 (0.406)	1.395 (1.099–1.769)	0.00614
CC	73.98 (0.135)	65.31 (0.117)	1.157 (0.810–1.651)	0.42264
CT	204.01 (0.371)	258.69 (0.462)	0.667 (0.524–0.848)	0.00095
AC	(0.000)	8.69 (0.016)		

Frequency distributions of haplotypes of *IL-21* polymorphisms. RA: Rheumatoid Arthritis; CS: control subjects. OR: Odds Ratio; 95% CI: 95% Confidence Intervals.

**Table 4 genes-14-00878-t004:** Association of clinical and serological parameters with *IL-21* and *IL-21R* polymorphisms in RA patients.

Polymorphisms	Clinical Activity and Serological Parameters					
Genotypes	DAS28-ESR Mean SD (*p*)	ESR Mean SD (*p*)	CRP Mean SD (*p*)	RF Mean SD (*p*)	CCP Mean SD (*p*)	IL-21 Mean SD (*p*)
rs2055979						
CC	3.803 ± 1.58	33.21 ± 17.87	19.93 ± 27.03	56.58 ± 27.87	162.6 ± 176.7	249.4 ± 543
CA	3.915 ± 1.39	35.38 ± 15.61	19.86 ± 25.44	62.59 ± 27.50	135.7 ± 173.9	181.6 ± 305
AA	4.090 ±1.41	36.69 ± 13.37	17.24 ± 23.62	63.71 ± 24.77	204.2 ± 231.5 **(0.0296) ***	279.9 ± 587
rs2221903						
TT	3.959 ± 1.39	35.88 ± 15.36	19.45 ± 27.18	63.84 ± 26.05 **(0.0292) ***	159.9 ± 184.6	215.4 ± 422
TC	3.843 ± 1.59	33.26 ± 16.60	19.00 ± 19.67	54.29 ± 28.79	153.4 ± 208.9	197.3 ± 404
CC	3.845 ± 0.96	31.50 ± 23.33	12.55 ± 12.94	65.37 ± 35.34	66.52 ± 82.89	1457 ± 1928
rs3093301						
TT	3.889 ± 1.40	35.39 ± 15.44	17.84 ± 19.56	59.14 ± 25.89	133.9 ± 159.3	196.4 ± 446
TA	3.855 ± 1.41	35.30 ± 16.84	18.71 ± 26.24	60.77 ± 28.95	167.6 ± 195.8	220.4 ± 440
AA	4.178 ± 1.60	34.27 ± 13.71	23.80 ± 33.19	67.96 ± 23.95	184.1 ± 232.1	268.7 ± 454

DAS28-ESR: Disease Activity Score 28-erythrocyte globular sedimentation rate; CRP: C-reactive protein; RF: rheumatoid factor; Anti-CCP: anti-cyclic citrullinated peptide antibodies. * Only significant *p*-Values are shown in bold. Test used Kruskal–Wallis/multiple comparison with Dunn’s correction.

**Table 5 genes-14-00878-t005:** Combined effect of *IL-21* (rs2055979/rs2221903) polymorphisms with clinical activity and serological parameters.

Combined Effect of *IL-21* Polymorphisms		Clinical Activity and Serological Parameters						
*IL-21* rs2055979	*IL-21* rs2221903	*n*	DAS28 Mean (CI) * *p* = 0.6852	ESR Mean (CI) * *p* = 0.4138	CRP Mean (CI) * *p* = 0.6852	RF Mean (CI) *** *p* = 0.0066**	CCP Mean (CI) * *p* = 0.1511	IL-21 Mean (CI) * *p* = 0.7160
AA	TT	62	4.090 (3.7–4.4)	36.69 (32–40)	17.24 (11–23)	63.71 (57–70)	204.2 (145–263)	279.9 (130–429)
CA	TC	42	3.932 (3.4–4.3)	35.77 (30–41)	18.22 (12–23)	51.62 * (42–60) ***p* = 0.0128**	151.7 (78–225)	155.5 (77–233)
CA	TT	106	3.908 (3.6–4.1)	35.22 (32–38)	20.51 (15–25)	66.93 * (62–71)	129.4 (101–156)	191.9 (129–254)
CC	TC	28	3.710 (2.9–4.4)	29.63 (23–36)	20.18 (11–28)	58.29 (47–69)	155.9 (92–219)	259.9 (41–478)
CC	TT	35	3.886 (3.3–4.4)	36.66 (29–43)	20.15 (9–30)	54.72 (44–64)	173.5 (107–239)	172 (62–218)
CC	CC	2	NA ^#^	NA ^#^	NA ^#^	NA ^#^	NA ^#^	NA ^#^

DAS28-ESR: Disease Activity Score 28-erythrocyte sedimentation rate; CRP: C-reactive protein; RF: rheumatoid factor; Anti-CCP: anti-cyclic citrullinated peptide antibodies. Data are presented as mean and Confidence Intervals. * Test used Kruskal–Wallis/multiple comparison with Dunn’s correction. The bold letters indicate significant differences. NA: Not applicable. ^#^ Due to a small *n,* this analysis could not be performed.

## Data Availability

All relevant data are included in the manuscript.

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
