# Peer review of "IL-21 (rs2055979 and rs2221903)/IL-21R (rs3093301) Polymorphism and High Levels of IL-21 Are Associated with Rheumatoid Arthritis in Mexican Patients"

_genes, 2023, doi:10.3390/genes14040878_

Round 1
Reviewer 1 Report
The authors investigated the association of IL-21/ IL-21R polymorphisms with RA and IL-21 serum levels in RA patients.
With good intentions, a lot of data are given. However, it seems that the crucial data, such as the association between IL-21/ IL-21R polymorphisms and IL-21 serum levels, are not properly introduced. Moreover, this is the main core of the title (Study of the polymorphisms IL -21 (rs2055979 and rs2221903) and IL -21R rs3093301 with levels of IL -21...) and the article (line 32 in abstract). As I said, it seems that this information is overlooked in the abstract and also in the conclusion part.
If the association of IL-21/ IL-21R polymorphisms with IL-21 serum levels is the core of the title and the authors did not find a significant association, this should be omitted from the title and the title of this article needs to be completely rephrased.
These non-significant results should be mentioned in the abstract and conclusion as well.
Author Response
Thanks for your comments. We agree with these observations. We have made some corrections in the manuscript, in the introduction section (lines 144-152) to explain the IL-21/IL-21R polymorphisms and IL-21 levels.
Likewise, we have restructured the title of the manuscript, the abstract and conclusions (abstract, lines 39-42) highlighting that there was no association of the polymorphisms with the levels of IL-21; but that there was an association between polymorphisms and RA; and on the other hand, elevated levels of IL-21 were observed in RA. We also highlight these findings in the conclusions section of the manuscript (lines 650-655).
Thank you so much for your comments.
Other changes in the manuscript according to some general aspects considered in their review were:
Information related to polymorphisms was included in the introduction section (lines 144-151).
In the methodology section, the exclusion criteria were included (lines 159-162); information on the calculation and analysis of the IL-21 ELISA (249-250). We moved up the paragraph that describes serological and inflammatory parameters (lines 207-215).
In the results section, we now include a graph of anti-CCP levels (figure 1). The IL-21 SNP and AR haplotype analysis, we moved them up, now they are 3.3 results (lines 406-417).
Information on IL-21 levels and disease activity are described in lines 447-450.
The conclusion was rewritten to make it more understandable and considering the findings of polymorphisms and levels of IL-21 in RA (lines 649-654).
Some changes in the manuscript for its improvement in the English language.
We believe that with your comments and the corrections made, the manuscript has improved substantially.
Reviewer 2 Report
1. What was the sensitivity (the lowest concentration that can be detected with acceptable accuracy) of the ELISA kit used. If it was 16 pg/ml, then the values below that cannot be considered reliable.
2. Likewise, what was the linear range for the assay? For example, if your linear range was 31.3-2,000 pg/mL, then values above 2,000 pg/mL can not be considered accurate. Also what was the EC50 value for the ELISA?
3. What was the exclusion criteria for the selected RA patients?
4. If possible, try to show anti-ccp ELISA data with a graph just like Figure 1.
Author Response
- Thanks for your comments and observations. The sensitivity indicated by the ELISA kit manufacturer is 16 pg/ml. However, we performed the calculation of the concentrations using a four-parameter logistic (4-PL) regression, which allows us to fit the curve with a wider range than the linear range of the standard curve. Due to this reason, some of the values obtained were below 16 pg/mL. It is also important to clarify that, when making the blank correction, some samples had an OD value less than 0, in these cases we consider a concentration = 0 pg/mL. It is also important to explain that when we performed the calculation of the optical densities, a correction was made by subtracting the OD value of the blank. In some samples, the result was an OD -0, and in these cases, we considered an IL-21 concentration of 0 pg/mL.
In addition, considering this observation by the reviewer, to be certain that our results are significant, we decided to perform the statistical analyzes between the study groups again, but omitting the values that were outside the ELISA range, which maintained the difference significant. This is now described in the methodology (lines 249 and 250).
- We appreciate your valuable questions. Regarding your first question, according to the standard concentrations indicated by the manufacturer, the linear range of the assay was from 31.3 to 2000 pg/ml. As we mentioned in the previous answer, we calculate the concentrations using a four parameter logistic regression (4PL), which allows us to fit the curve with a wider range than the linear range of the standard curve. Due to this reason, some of the values were higher than 2000 pg/mL.
Regarding your second question, the EC50 value was 905 pg/mL.
The Best-Fit Values of 4PL were: Top = 1896 pg/mL; bottom = 1.07 pg/mL; EC50 = 905 pg/mL; with a 95% CI, and a R-square = 0.9987.
- Thanks for this question. The exclusion criteria were coexistence of other autoimmune diseases, uncontrolled sys-temic diseases or malignancies, treatment with corticosteroids at doses greater than 15 mg per week, pregnancy, and presence of inflammatory or infectious diseases at the time of blood sample collection. These criteria are now included in the methodology section (lines 159 - 162).
- We appreciate this suggestion. This graph is now included in the manuscript as figure 1.
Other changes in the manuscript according to some general aspects considered in their review were:
Information related to polymorphisms was included in the introduction section (lines 144-151).
In the methodology section, the exclusion criteria were included (lines 159-162); information on the calculation and analysis of the IL-21 ELISA (249-250). We moved up the paragraph that describes serological and inflammatory parameters (lines 207-215).
In the results section, we now include a graph of anti-CCP levels (figure 1). The IL-21 SNP and AR haplotype analysis, we moved them up, now they are 3.3 results (lines 406-417).
Information on IL-21 levels and disease activity are described in lines 447-450.
The conclusion was rewritten to make it more understandable and considering the findings of polymorphisms and levels of IL-21 in RA (lines 649-654).
Some changes in the manuscript for its improvement in the English language.
We believe that with your comments and the corrections made, the manuscript has improved substantially.
Round 2
Reviewer 1 Report
The authors improved their manuscript.